# Current-induced switching of proximity-induced ferromagnetic surface states in a topological insulator

Masataka Mogi [1,2,5✉], Kenji Yasuda [1,5], Reika Fujimura[1], Ryutaro Yoshimi [2], Naoki Ogawa[1,2], Atsushi Tsukazaki [3], Minoru Kawamura [2], Kei S. Takahashi[2], Masashi Kawasaki[1,2] & Yoshinori Tokura [1,2,4✉]

Electrical manipulation of magnetization could be an essential function for energy-efficient spintronics technology. A magnetic topological insulator, possessing a magnetically gapped surface state with spin-polarized electrons, not only exhibits exotic topological phases relevant to the quantum anomalous Hall state but also enables the electrical control of its magnetic state at the surface. Here, we demonstrate efficient current-induced switching of the surface ferromagnetism in hetero-bilayers consisting of the topological insulator $(Bi_{1-x}Sb_x)_2Te_3$ and the ferromagnetic insulator $Cr_2Ge_2Te_6$, where the proximity-induced ferromagnetic surface states play two roles: efficient charge-to-spin current conversion and emergence of large anomalous Hall effect. The sign reversal of the surface ferromagnetic states with current injection is clearly observed, accompanying the nearly full magnetization reversal in the adjacent insulating $Cr_2Ge_2Te_6$ layer of an optimal thickness range. The present results may facilitate an electrical control of dissipationless topological-current circuits.

[1] Department of Applied Physics and Quantum Phase Electronics Center (QPEC), University of Tokyo, Bunkyo-ku, Tokyo, Japan. [2] RIKEN Center for Emergent Matter Science (CEMS), Wako, Saitama, Japan. [3] Institute for Materials Research, Tohoku University, Sendai, Miyagi, Japan. [4] Tokyo College, University of Tokyo, Bunkyo-ku, Tokyo, Japan. [5]Present address: Department of Physics, Massachusetts Institute of Technology, Cambridge, MA, USA. ✉email: mogi@mit.edu; tokura@riken.jp

Spin-polarized surface electronic states of three-dimensional topological insulators (TIs) offer novel physical properties, being potentially applicable to future low-power-consumption electronics/spintronics and topological quantum computation[1]. One of the representative features is the emergence of anomalous Hall conductance in the gapped surface state when magnetization perpendicular to the surface is induced by the incorporation of magnetic elements or proximity coupling with a ferromagnetic (FM) layer on the TI[2–8]. This magnetic TI exhibits exotic magnetic insulating phases, such as a quantum anomalous Hall (QAH) insulator and an axion insulator. In particular, the QAH state provides a research arena based on the non-dissipative chiral edge conduction, whose direction is determined by the magnetization direction. By controlling the magnetization direction, a topological invariant of the Chern number $C$ in the QAH conditions can correspondingly be controlled, as manifested by the switching of the conduction direction ($C = 1$ or $-1$) and on/off switching ($|C| = 1$ or $0$) of the chiral edge channel[4–6].

Besides external magnetic fields, an electric current injection can also control the magnetization directions. Owing to the spin-momentum locking of the electrons at the TI surface state[9,10], the flow of electrons produces nonequilibrium spin accumulation, which exerts spin torques on a FM layer adjacent to the TI surface[11,12]. The current-induced switching of the perpendicular magnetization in the FM layer in conjunction with a TI has been demonstrated in a highly efficient manner with lower critical current densities than those in FM-metal/heavy-metal heterostructures[13–19]. Moreover, the spin accumulation may also enable the electrical manipulation of the TI surface ferromagnetism that originates from the FM proximity coupling at the FM-layer/TI interface. However, in the intensively studied FM-metal/TI systems, the current at the TI surface state is mostly shunted by the FM-metal layer. Hence, the replacement of the FM metal with a FM insulator (FMI) would be effective to control the magnetization with lower electric current[20–22], which is also required to observe the QAH states with nontrivial Chern numbers[8]. It has been demonstrated that the magnetization in the FMI layer can be partly switched by current excitation in magnetically doped TI[13,14] and FMI/TI heterostructures[20], but the switching ratio of the magnetization has been limited to <50%. Such a low ratio, which likely originates from the inhomogeneous nature of doping-induced ferromagnetism and weak magnetic coupling at the interface of FMI/TI, respectively, results in the formation of multi-domain states after each switching operation and causes difficulty in the control of the topological invariant. Therefore, it is expected that the choice of a suitable FMI with high crystallinity and strong interfacial coupling can achieve a high switching ratio.

Here, we demonstrate the nearly full magnetization switching in an all-telluride-based intrinsic FMI ($Cr_2Ge_2Te_6$; CGT)/TI heterostructure (Fig. 1a). The layered FMI compound of CGT[23,24] has recently been found to provide strong proximity coupling with the TI surface state, as exemplified by the observation of large anomalous Hall effect (AHE) originating from a prominent Berry curvature near the exchange gap formed in the proximity-coupled surface state[25] (Fig. 1b). Since the efficient spin transfer to the FMI layer requires strong couplings with well-ordered interfaces between the TI surface state and the magnetic moments in the FMI layer[26,27], the CGT/TI heterostructures may be of great advantage for highly efficient switching. Through the optimization of the CGT thickness for the switching ratio and efficiency, the magnetization of the CGT layer can be almost fully switched by the in-plane current injection on the TI layer as low as 2–4 A cm$^{-1}$ in the CGT thickness range of 3–5 nm (corresponding to four to seven CGT monolayers). Furthermore, high switching current efficiencies irrespective of the Fermi level ($E_F$) indicate that the spin torques are dominantly generated from the

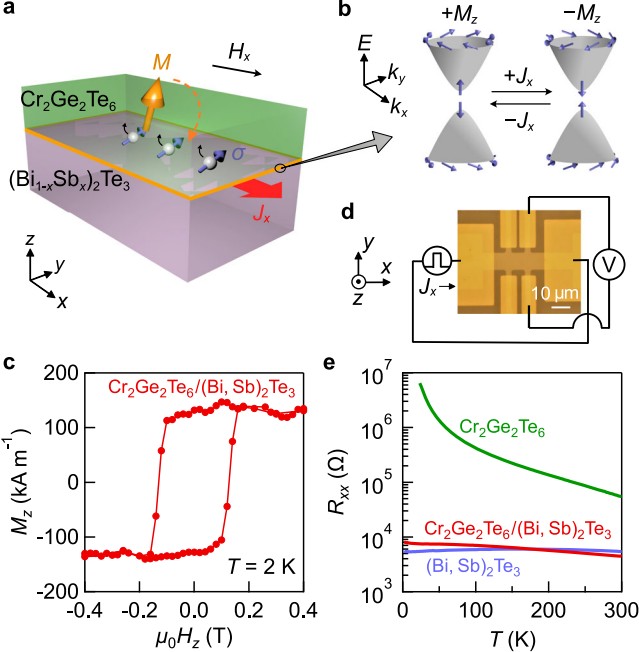

**Fig. 1 CGT/BST bilayer device. a** Schematic illustration of magnetization switching in $Cr_2Ge_2Te_6/(Bi_{1-x}Sb_x)_2Te_3$ (CGT/BST) bilayer. The magnetization $M$ in CGT is reversed by current injection along $x$-direction ($J_x$) under an in-plane magnetic field ($H_x$) parallel to $J_x$ via the spin–orbit torque from the spin ($\sigma$)-polarized surface state conduction. The perpendicular component of the magnetization is probed by AHE. **b** Schematic illustration of the electronic structures of surface states gapped by the magnetic proximity coupling. The magnetization switching in the CGT layer corresponds to the switching of the sign of the magnetic gap at the proximity-coupled surface states. **c** Out-of-plane magnetic field $\mu_0H_z$ dependence of the magnetization $M_z$ in the CGT (12 nm)/BST ($x = 0.5$, 6 nm) bilayer at 2 K. **d** Optical microscope image of a Hall bar device with an illustration of the measurement setup. **e** Longitudinal resistance $R_{xx}$ as a function of temperature $T$ for the CGT single-layer (12 nm; green), BST($x = 0.5$) single-layer (6 nm; purple), and CGT (12 nm)/BST ($x = 0.5$, 6 nm) bilayer (red) films.

TI surface state rather than its bulk state. Such a FMI/TI heterostructure is highly suitable for the current-induced control of the surface ferromagnetism of TI.

## Results

**Heterostructure and device characterization.** We grew CGT/$(Bi_{1-x}Sb_x)_2Te_3$ (CGT/BST) bilayer films (Fig. 1a) on InP(111) substrates by molecular beam epitaxy (MBE; see "Methods" section). To examine the $E_F$ position dependence, the Bi/Sb ratio ($x$) was systematically controlled in the BST layer. The previous study[25] on the CGT/BST heterostructure, which was prepared in the same way as in the present study, has proven the high crystal quality and well-ordered, sharp interfaces due to van der Waals bonding, as well as negligible atomic interdiffusion by cooperatively using x-ray diffraction, depth-sensitive x-ray/neutron reflectometry, and cross-sectional scanning transmission electron microscopy/energy-dispersive x-ray spectroscopy (Supplementary Note 1). The magnetization of a MBE-grown CGT (12 nm)/BST (6 nm) structure (the Curie temperature: $T_C \sim 80$ K) at a temperature of $T = 2$ K (Fig. 1c) shows the hysteresis loop with the out-of-plane easy axis, which is nearly identical with the property of the MBE-grown CGT single layer itself[24]. For the electrical transport measurements and magnetization switching characterizations, we fabricated Hall bars with 10 μm in width (Fig. 1d).

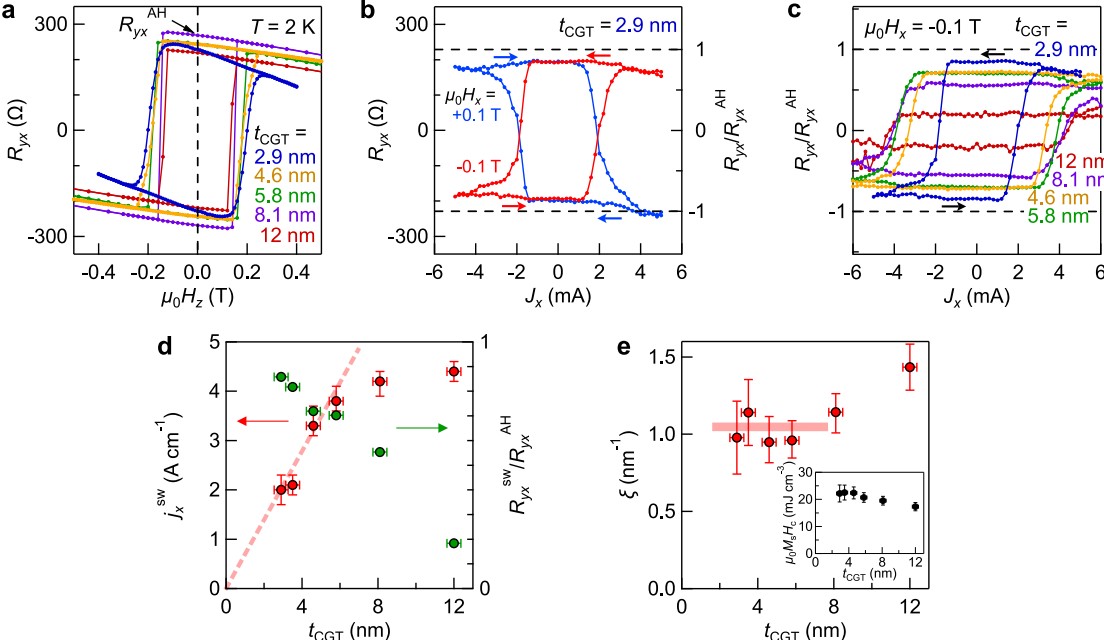

**Fig. 2 Dependence of proximity-induced AHE and current-induced magnetization switching on the CGT layer thickness. a** $\mu_0 H_z$ dependence of the Hall resistance $R_{yx}$ in the CGT/BST ($x = 0.5$, 6 nm) bilayer devices with various CGT thickness ($t_{CGT} = 2.9$, 4.6, 5.8, and 12 nm) at 2 K. $R_{yx}^{AH}$ is the remnant value of $R_{yx}$ at $\mu_0 H_z = 0$ T. **b** Magnetization switching with current pulses $J_x$, as tracked by the variation of Hall resistance $R_{yx}$ (left axis) of the CGT ($t_{CGT} = 2.9$ nm)/BST ($x = 0.5$, 6 nm) device under in-plane magnetic fields $\mu_0 H_x = +0.1$ T (blue) and $-0.1$ T (red) at 2 K. Right axis is the switching ratio defined as $R_{yx}/R_{yx}^{AH}$. The broken lines represent $R_{yx}/R_{yx}^{AH} = 1$ for the full switching of magnetization. **c** Magnetization switching in the CGT/BST devices with various $t_{CGT}$ (=2.9, 4.6, 5.8, 8.1, and 12 nm) under $\mu_0 H_x = -0.1$ T. **d, e** The $t_{CGT}$ dependence of the switching current $J_x^{sw}$ (left axis) and the switching ratio of $R_{yx}^{sw}/R_{yx}^{AH}$ (right axis) (**d**), and the coefficient $\xi$ [$=2e\mu_0 M_s H_c t_{CGT}/(\hbar j_x^{sw})$] representing the efficiency of current-induced magnetization reversal (**e**). Note that $\xi$ for $t_{CGT} = 8$ and 12 nm increases up to ~1.4 nm$^{-1}$, where the Joule heating seemingly improves the efficiency. The inset to **e** shows the $t_{CGT}$ dependence of the product of the coercive field $\mu_0 H_c$ and the spontaneous magnetization $M_s$. The horizontal error bars represent the film roughness determined by x-ray reflectivity measurements. The vertical ones for **d** and **e** represent the measurement uncertainties. The red broken (**d**) and solid (**e**) lines are the guides to the eyes.

Figure 1e shows the $T$ dependence of sheet resistance $R_{xx}$ for the CGT single-layer (12 nm), BST ($x = 0.5$, 6 nm) single-layer, and CGT (12 nm)/BST ($x = 0.5$, 6 nm) bilayer films. Whereas the CGT single-layer film exhibits a highly insulating behavior (>1 M$\Omega$ <50 K), the resistance of the CGT/BST bilayer film is comparable to that of the BST single-layer film. The electrical conductivity in the TI layer depends on the $E_F$, yet the dominant conduction channel is still in the TI layer for all the heterostructures (Supplementary Fig. 2).

**Proximity-induced anomalous Hall effect**. We first discuss the CGT layer thickness ($t_{CGT}$) dependence of the AHE for six samples of the CGT/BST($x = 0.5$, 6 nm) bilayers. This composition $x = 0.5$ is the optimum value in the bilayer system for the observation of the largest AHE as discussed later. We note that the optimum $x$ value is different from the charge-neutrality point value ($x \sim 0.8$) in BST single-layer films[3–5,8] to suppress a possible charge transfer at the CGT/BST interface. Figure 2a shows the Hall resistance $R_{yx}$ of typical four samples, exhibiting comparable AHE responses. In addition, the coercive field of the hysteresis loops is consistent with the magnetization hysteresis curve of the CGT layer ($t_{CGT} = 12$ nm; Fig. 1c). The value of remnant anomalous Hall resistance $R_{yx}^{AH}$ at zero magnetic field (~250 $\Omega$) is nearly constant irrespective of $t_{CGT}$ ($2.9 \leq t_{CGT} \leq 12$ nm), reflecting that the Berry curvature generated at the magnetically gapped TI surface state.

**Current-induced magnetization switching**. In Fig. 2b, we present the current-induced magnetization switching of the CGT

(2.9 nm)/BST ($x = 0.5$, 6 nm) bilayer. To perform the perpendicular magnetization switching[28–30], 100-$\mu$s duration current pulses ($J_x$) were injected under in-plane magnetic fields ($\mu_0 H_x = \pm 0.1$ T) much smaller than the anisotropy fields ($H_K \sim 0.9$ T; see Supplementary Note 2). After every current pulse injection, $R_{yx}$ was measured with a much smaller probe current of $J_x = 10$ $\mu$A to elucidate the magnetization direction of the CGT layer and the proximity-induced surface ferromagnetism via Hall measurement. As shown in Fig. 2b, when the amplitude of the injection current pulse exceeds the switching threshold current $J_x^{sw} \sim 2.0$ mA, the sign of the Hall resistance is reversed. As expected from the antidamping-like spin–orbit torque switching[28–30], the switching polarity is reversed when we reverse the direction of the in-plane magnetic field ($H_x$ blue curve) $\rightarrow -H_x$ (red curve)). The change of Hall resistance $R_{yx}^{sw} = |R_{yx}(J_x \rightarrow +0,\ H_x < 0) - R_{yx}(J_x \rightarrow -0,\ H_x < 0)|/2 = 200\ \Omega$ is comparable to the remnant Hall resistance $R_{yx}^{AH} = 228\ \Omega$, in which the ratio $R_{yx}^{sw}/R_{yx}^{AH} \sim 0.88$ corresponds to the reversed magnetization ratio in the CGT. This large value means the realization of the nearly full magnetization switching, resulting in the switching of the FM surface states and the topological spin structures in the surface state (Fig. 1b).

**Cr₂Ge₂Te₆ thickness dependence of the magnetization switching**. Having established the current-induced magnetization switching, we next examine the $t_{CGT}$ dependence of $J_x^{sw}$ to evaluate the switching ratio and efficiency. As shown in Fig. 2c, the $J_x^{sw}$ increases with increasing $t_{CGT}$, while the switching volume fraction measured by $R_{yx}^{sw}/R_{yx}^{AH}$ decreases. A linear increase of $J_x^{sw}$ in Fig. 2d saturates at $6 < t_{CGT} < 12$ nm. Furthermore, $R_{yx}^{sw}/R_{yx}^{AH}$

drastically decreases at $6 < t_{CGT} < 12$ nm. The $t_{CGT}$-linear relation for $t_{CGT} < 6$ nm is attributed to that the spin torques required for the switching linearly increase with the spontaneous magnetization per sample area, $M_s t_{CGT}$. On the other hand, for $t_{CGT} = 8$ and 12 nm, the large $M_s t_{CGT}$ requires a large $J_x^{sw}$, resulting in a thermal instability of magnetization due to the Joule heating which perhaps not only assists the switching, but also reduces $R_{yx}^{sw}/R_{yx}^{AH}$ due to the multi-domain formation (see Supplementary Note 4 for the estimation of heating effect).

To quantitatively compare the $t_{CGT}$-dependent switching behavior, we define a coefficient parameter $\xi = 2e\mu_0 M_s H_c t_{CGT}/(\hbar j_x^{sw})$ (refs. [12,30]). Here $e$ is the elementary charge, $\hbar$ is the reduced Planck's constant, $\mu_0 H_c$ is the coercive field of the CGT layer, and $j_x^{sw} = J_x^{sw}/W$ ($W = 10$ μm: the width of the Hall bars) is the sheet switching current density. This coefficient describes how efficiently the surface spin accumulation is absorbed in the FMI layer via the charge-to-spin current conversion ($J_s = \xi j_x$), where the resulting antidamping-like spin–orbit torque magnitude (in a unit moment) is described by $\tau_{AD} = \hbar J_s/(2eM_s t_{CGT})$. Since the switching is likely driven by current-induced domain nucleation and subsequent domain wall motion processes[30] due to the large-scale Hall bar (Fig. 1d) compared with the magnetic domain size (see Supplementary Fig. 6 for the observed magnetic domain structure), the spin–orbit torque acts as the domain wall depinning field, which is assigned to the coercive field of the CGT layer ($\mu_0 H_c$) at the switching threshold current injection[30]. As shown in Fig. 2e, we observe $\xi$ for $t_{CGT} < 6$ nm takes a roughly constant value of $\xi \sim 1.0$ nm$^{-1}$, being consistent with the linear $J_x^{sw}$–$t_{CGT}$ relation shown in Fig. 2d. Furthermore, this value obtained without any correction of current distribution in the TI layer owing to the insulating FM layer is comparable to that derived in the previous spin–torque FM resonance experiment on the FM-metal layer in proximity to the TI BST[12]. Incidentally, to more quantitatively estimate the spin-charge conversion efficiency, we tried to utilize the prevailing method to use the second harmonic Hall measurement[31,32]. The analysis based on it, however, gave an unphysically large efficiency value, i.e., ~1000 nm$^{-1}$. Such a large nonlinear Hall signal is perhaps due to a magnon scattering of the spin-momentum-locked surface Dirac electrons of TI[14] (see Supplementary Note 6 for details).

**Fermi-level position dependence of the switching efficiency.** We turn to $E_F$ position dependence of magnetization switching for the evaluation of the surface state contribution. Figure 3a shows the AHE in CGT (3.5 nm)/BST (6 nm) having different $x$ with the fixed thicknesses of each layer. Judging from the sign of the ordinary Hall term at high magnetic fields, the dominant carrier type is systematically controlled with increasing $x$ from electron type for $x = 0$, 0.3, and 0.5 to hole type for $x = 0.7$ and 1. At the $x = 0.5$, $R_{yx}^{AH}$ is maximized, indicating that $E_F$ is closest to the magnetic gap where the Berry curvature contribution is the largest[2,3]. In addition, the sign of AHE is reversed, while reducing its magnitude, from positive to negative for $0.5 < x < 0.7$, reflecting the sign change of the Berry curvature. Such a sign change has been observed also in magnetically doped TI heterostructures[33], where an additional anomalous Hall conductivity with the opposite sign is generated from Rashba-split bulk valence bands due to the broken inversion symmetry by the heterostructure. Hence, both the surface and bulk states are likely to contribute to the electrical conduction in p-type BST ($x = 0.7$ and 1). We thus observed several characteristic behaviors in the current-induced magnetization switching in Fig. 3b. First, its switching polarity change ($R_{yx}$ versus $J_x$) between $x = 0.5$ and 0.7 coincides with that of AHE ($R_{yx}$ versus $H_z$), indicating the same spin–torque directions irrespective of the carrier types. Second, $R_{yx}^{sw}/R_{yx}^{AH}$ is

nearly constant against the variation of $x$. Third, $J_x^{sw}$, on the other hand, varies with $x$, where the minimum 2 mA is observed for $x = 0.5$, while both $x = 0$ and 1 require the larger $J_x^{sw}$ of 5 mA, implying the switching efficiency depends on the $E_F$.

To clarify the relationship between the switching efficiency and the $E_F$ position, we compare the $x$ dependence of $\xi$ and $R_{yx}^{AH}$ in Fig. 3c. Both are enhanced at $x = 0.5$ in which the bulk conduction is mostly suppressed with $E_F$ being close to the magnetic gap of the TI surface state. The large $R_{yx}^{AH}$ (solid blue circles) comes from the prominent Berry curvature generated around the gap. Moreover, the broad peak of $\xi$ $x = 0.5$ implies that the spin-polarized surface state plays a dominant role in the magnetization switching because the bulk contribution present at $x = 0$ and 1 does not increase $\xi$. Thus, $E_F$ tuning is advantageous for improving the switching efficiency, as well as for maximizing the AHE while the almost full magnetization switching is accomplished for all the samples (Fig. 3b). Note that despite the surface state is magnetically gapped by the proximity coupling with the FMI layer, suppression of the charge-to-spin current conversion efficiency in the nearly charge-neutral samples is not observed, possibly due to still unprecise tuning of the Fermi level into the middle of the exchange gap or to the spatially inhomogeneous gap opening.

## Discussions
Finally, we argue that these magnetization switching features are consistent with current-induced dynamics of the spin-momentum-locked TI surface state as the dominant source of the spin torques[11–19]. First, possibilities for the source of the spin torques other than the topological surface states may include the spin Hall effect from the bulk bands[34] and the inversion symmetry breaking of the hetero-interface, where a vertical electric field at the interface can induce Rashba spin splitting in the bulk states[35]. However, these scenarios cannot account for the present observation that the efficiency is increased when the $E_F$ is within the TI bulk gap rather than in the bulk states. Second, the accumulated spin direction is irrespective of the carrier types. The Fermi circle with the opposite spin helicities for n-type and p-type (Fig. 3c) suggests that the direction of spins would be opposite for the carrier types. However, in consideration of the Fermi surface response to the electric current or field, the accumulated spin directions are the same as elucidated in the following. When an electric field ($+E_x$) is applied, the shift of the Fermi circle has the same direction irrespective of the carrier type: $k_x \rightarrow k_x - \frac{eE_x\tau}{\hbar} = k_x - \delta k$. Then, as shown in Fig. 3e, if $E_F > E_{DP} = 0$, the Fermi circle shift increases the population of electrons for the $-k_x$ branch, while reducing that for the $+k_x$ branch. On the other hand, if $E_F < E_{DP}$, the Fermi circle shift reduces the population of electrons for the $-k_x$ branch, while increasing that for the $+k_x$ branch (Fig. 3f). Hence, the increased spin populations for the n-type and p-type have the same direction $\hat{\sigma} = +\hat{y}$ under $+E_x$. Since the antidamping effective field, which is given by $\hat{\sigma} \times \hat{m}$, does not depend on the momentum but the spin direction, the spin–orbit torque direction is not changed by the carrier types. Third, the magnetization switching direction itself is consistent with the spin-momentum-locked surface state of BST by using a macrospin model[29]: the effective field originating from antidamping torques $\boldsymbol{\tau}_{AD} = \tau_{AD}(\hat{m} \times (\hat{\sigma} \times \hat{m}))$, where $\tau_{AD} > 0$, is described by $\mathbf{H}_{AD} = (\tau_{AD}/\mu_0)\hat{\sigma} \times \hat{m}$. Suppose the magnetic moment is $\hat{m} = +\hat{z}$, the effective field $\mathbf{H}_{AD} || +\hat{x}$ under the current pulse of $j_x > 0$ (i.e., $\hat{\sigma} = +\hat{y}$ due to the spin-momentum locking nature of BST), which rotates the magnetization via magnetic damping as shown in Fig. 1a. When the magnetic moment is slightly tilted to $+\hat{x}$ ($-\hat{x}$) by an in-plane magnetic field, the $\hat{m} = -\hat{z}$ ($+\hat{z}$) state is favored under the current pulse injection, which well describes the observed behaviors shown in Fig. 2b. Note that once the reversed magnetic

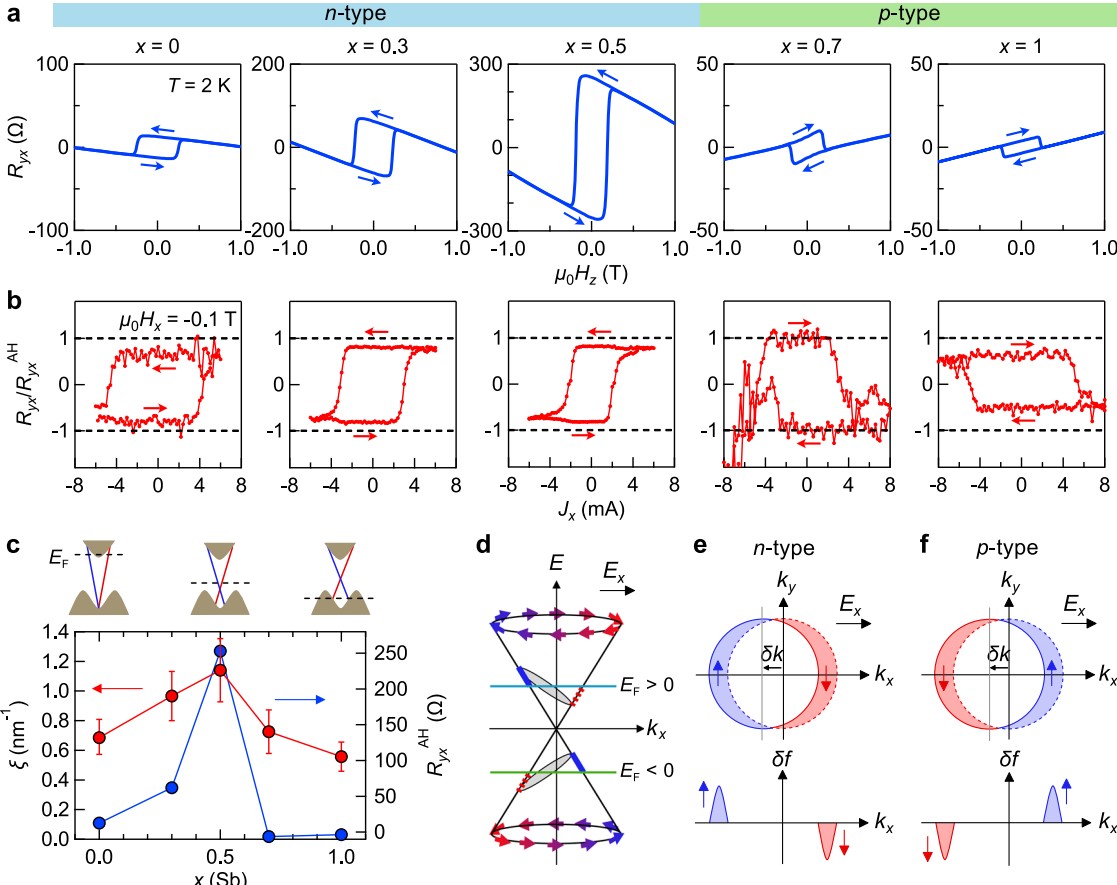

**Fig. 3 Fermi-level dependence of magnetization switching. a** AHE showing $R_{yx}$ versus $\mu_0H_z$ for CGT ($t_{CGT}$ = 3.5 nm)/(Bi$_{1-x}$Sb$_x$)$_2$Te$_3$ (6 nm) with various $x$ (=0, 0.3, 0.5, 0.7, and 1). The samples are categorized to be n-type ($x$ = 0, 0.3, and 0.5) or p-type ($x$ = 0.7 and 1). **b** Magnetization switching under in-plane magnetic fields $\mu_0H_x$ = −0.1 T. **c** Sb content ($x$) dependence of $\xi$ (solid red circle) and $R_{yx}^{AH}$ (solid blue circle). Simplified schematics of the band structures with the $E_F$ positions are depicted above the panel. The vertical error bars represent the measurement uncertainties. **d** Illustration of the energy dispersion of the TI surface state. **e**, **f** The spin accumulation driven by a shift of the Fermi surface (top) and the difference in the Fermi distribution for electrons $\delta f$ under an electric field ($E_x$; bottom) for the n-type (**e**) and p-type (**f**) TI.

domains are nucleated by the above macrospin model mechanism, the domains may be expanded toward the single domain state more efficiently than the macrospin rotation[29,30].

In conclusion, by the current excitation at the TI surface state, we have successfully demonstrated the nearly full switching of the FM surface states in the TI layer proximity-coupled to the insulating CGT. The systematic CGT thickness $t_{CGT}$ dependence of the switching current reveals that the $t_{CGT}$ should be <6 nm to realize the full switching. Our results indicate the compatibility of the large proximity-induced AHE and the efficient magnetization switching, paving a way to electrically manipulate topological quantum states. For instance, thick FMI/TI/thin FMI sandwich structures[8,25] with different switching currents (Fig. 2d) for the top and bottom TI surfaces would allow selective controls of the magnetic layers, namely electrical switching between the QAH insulator (parallel magnetization) and axion insulator (anti-parallel magnetization) states[5,6]. Whereas such topological states are surface insulating and would not directly contribute to the spin–torque generation, an additional electrostatic gating capability to control the Fermi level[36] or a current-driven breakdown of the QAH state during a current pulse injection[37–39] could retrieve the spin-polarized surface transport. This allows linking the spintronic functionalities to the topological quantum states, opening a new avenue to unprecedented control of dissipationless topological current devices.

## Methods

**Film growth and characterization.** The CGT (top)/BST (bottom) films were grown by MBE on semi-insulating InP substrates using standard Knudsen cells in a MBE chamber under a vacuum condition (~1 × 10$^{-7}$ Pa). The growth temperatures for the CGT layers and the BST layers were 180 °C and 200 °C, respectively[24,25]. Taking out the films from the MBE chamber, the AlO$_x$ capping layer (~5 nm) was immediately deposited by atomic layer deposition at room temperature. The crystal structures and thicknesses of the respective layers were confirmed by x-ray diffraction and reflectivity measurements, respectively[25].

**Device fabrication.** The films were patterned into Hall bars with 10 μm in width and 30 μm in length by using photolithography and chemical etching, with H$_2$O$_2$–H$_3$PO$_4$–H$_2$O and HCl–H$_2$O mixtures. The electrodes were made of Ti (5 nm)/Au (45 nm) deposited by electron beam evaporation.

**Electrical transport measurements.** The electrical transport measurements of the Hall bars were performed in a Quantum Design PPMS (2 K, 9 T). The current value for the resistivity and Hall effect measurements was 10 μA.

**Magnetization switching measurements.** The pulse current with varying pulse amplitudes between ±8 mA was injected into the Hall bar by using a current source (Keithley Model 6221). After the injection, the Hall resistance was subsequently measured with a voltmeter (Keithley Model 2182A) under a low probe current of 10 μA.

## Data availability

All relevant data within this paper are available from the authors upon reasonable request. Source data are provided with this paper.

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

## Acknowledgements

This research project was partly supported by JSPS/MEXT Grant-in-Aid for Scientific Research (Nos. 15H05853, 15H05867, 17J03179, 18H04229, and 18H01155) and JST CREST (No. JPMJCR16F1).

## Author contributions

M.M. and K.Y. planned the experiments. M.M. and R.F. fabricated the samples with the help of R.Y., A.T., M. Kawamura, K.S.T., and M. Kawasaki. M.M. and R.F. performed the transport and magnetization switching measurements, and analysed the data with the help of K.Y. and R.Y. K.Y. performed the magnetic force microscopy measurements. M.M., K.Y., R.F., N.O., A.T., and Y.T. discussed the results. M.M., A.T., and Y.T. wrote the manuscript with inputs from all the other authors. Y.T. and M. Kawasaki conceived and supervised the project.

## Competing interests

The authors declare no competing interests.
