## [Peer Review File · Nature Communications]

Editorial Note: Parts of this peer review file have been redacted as indicated to remove third-party material where no permission to publish could be obtained. Figure R8 on page 23 in this peer review file is reproduced with permission from American Physical Society.

REVIEWER COMMENTS

Reviewer #1 (Remarks to the Author):

Mogi and coworkers report current-induced switching of a ferromagnetic insulator in interfacial contact with a topological insulator (TI). Switching occurs presumably due to the spin-polarized surface states interacting with the magnetization of the neighboring FM insulator. They study the FM insulator layer thickness dependence and the TI composition dependence, which spans from the electron dominated to the hole dominated transport by tuning the Fermi energy. They convincingly show that this system is a very good candidate for current-induced magnetization manipulation in TI-based devices. Moreover, the systematic studies shed light on the charge-spin conversion mechanism being the surface states rather than the bulk. Overall, I believe that this is an essential work for the spintronics field and beyond. Therefore, I would recommend this work for publication in Nature Communications if the authors can address the following questions/issues.

-The authors quantify the switching efficiency by considering the threshold current and the coercivity of the films, together with other relevant parameters. However, based on their data, there are two potential problems with this approach. First, it ignores the magnetic anisotropy, which plays an important role in the switching process even though it is mediated by domain nucleation and propagation (see, e.g., PRB 89 214419 2014). This is because a fixed in-plane bias field will tilt the magnetization to a different angle depending on the perpendicular magnetic anisotropy. The larger the tilt angle is, the easier the switching will be. Therefore, the comparison in Fig.2e is valid only if the perpendicular anisotropy is comparable for different CGT thicknesses as the in-plane bias field was constant for all measurements. Second, the domain sizes and domain wall propagation properties will also influence the critical current density to achieve full switching. It is evident in Fig. 2a that these properties are different for the different CGT thicknesses. Thinner CGT samples tend to have more sheared loops, evidence for small domains, and significant pinning. I would recommend the authors perform a quantitative analysis of the current-induced damping-like torque for the different samples to compare the spin-torque efficiencies. This could be easily done by, e.g., harmonic Hall voltage analysis widely used in the metallic ferromagnetic heterostructures (see, e.g., Nat. Nano. 8 587 2013, Nat Mater 12 240 2013).

-Following up on the above comment, I recommend the authors report the perpendicular magnetic anisotropy estimates for the different samples used in this study.

-On page 6, the authors claim that for thicker CGT samples, the reduced switching might be due to multidomain formation. This can be easily tested with an out-of-plane field and an ac current injection of the same amplitude, which would reproduce the same Joule heating scenario.

-The explanation of the data of Fig.3 on page 8 is somewhat superficial (starting with the sentence "... These features are consistent with..."). The authors should explain more clearly how they arrive at the conclusion that the TI surface state is the dominant spin current source. They should clarify in a more intuitive way why the torque direction remains the same when the carrier type changes. Also, I believe that their explanation of torques on the domain walls is oversimplified. I would suggest the authors consider a simpler picture of the macrospin model and the action of the torque on the magnetization and not on the domain walls. Although it is clear that the switching is mediated by domain nucleation and propagation, the initial domain would still need to be nucleated following a macrospin model.

-Typo on page 4. The Hall bar width should be 10 μm and not 10 mm.

-The optical micrograph in Fig.1d could be improved. A higher resolution/contrast version would be better.

-There is a misleading reference. Ref#15 reports full switching in a TI/FM conductor heterostructure. The authors have referred to <50% switching in a TI/FM insulator structure while citing this work. The discussion on page 3 should be rephrased.

-The authors should estimate current density instead of stating the current itself, which depends on the device geometry.

Reviewer #2 (Remarks to the Author):

In the manuscript by M. Mogi et al., the authors studied the current-induced switching and spin-orbit torque (SOT) in the topological-insulator $(\text{Bi}_{1-x}\text{Sbx})_2\text{Te}_3$ /ferromagnetic-insulator $\text{Cr}_2\text{Ge}_2\text{Te}_6$ bilayer structure. Because of the high crystallinity and strong interfacial coupling in the bilayer, a high switching ratio (~ 1) is achieved. The authors also reported the $\text{Cr}_2\text{Ge}_2\text{Te}_6$ (CGT) layer thickness-dependent and $(\text{Bi}_{1-x}\text{Sbx})_2\text{Te}_3$ layer composition-dependent switching behaviors in this manuscript. While these results are interesting, given the abundant reports on the topological insulator (TI) -based SOT switching in the field, I don't feel this manuscript stands out as a breakthrough-type of work in this research field in terms of novelty. The manuscript should be further improved in order to be considered for the journal of Nature Communications:

(1) Throughout the manuscript, the authors mainly used the current-induced switching results to study the SOT in the BST/CGT bilayer structure. That is the main weakness of the work. We know that besides the switching behavior, there are several electrical-transport, magneto-optical, or microwave-induced methods to calibrate the SOT strength in the system. For example, the spin-torque ferromagnetic resonance, the hysteresis-shifting method in the presence of in-plane field and current, magneto-optical Kerr effect, etc. The authors should use one or more such methods to calibrate the SOT strength in their system, in order to have a better understanding of the SOT efficiency. The formula used by the authors (page 6, line 134) has certain constraint for describing the real SOT efficiency in the system. For example, the coercive field itself depends on many parameters, and it can also be influenced by the current applied through the film (by e.g., heating). So, this coefficient may not be the best formula to derive the SOT efficiency. I would suggest the authors use at least one of the above-mentioned methods to carefully calibrate the SOT in the system, so that the manuscript will be more comprehensive.

(2) Another point the authors will need to address is that the studied BST/CGT bilayer structure is very close to the magnetic-TI/TI bilayer structure. We know that the Cr dopants have quite good solubility in the BST material, so during the growth of the BST/CGT bilayer, there could be possibility that the Cr elements in the CGT are diffused into the bottom BST layer. In this case, the studied BST/CGT bilayer is not much different from the previously studied magnetic-TI/TI bilayer structures. I suggest the authors to provide more evidence that the SOT is switching the CGT layer, not the diffused Cr dopants into the TI or the formed Cr-doped TI spacer layer at the interface between CGT and BST. A thorough scanning TEM and elemental mapping could be helpful in addressing this issue.

(3) In the first two paragraphs, or the introduction part, the authors have addressed that the BST/CGT could be useful in studying the quantum anomalous Hall effect (QAHE), axion insulators, and SOT. While these objects all involved the topological materials, the mechanisms are quite different. For example, in the SOT study, the TI surface states are conducting while in the QAHE, the TI surface is

gapped and only the edge states participate in the electrical transport. It will be good if the authors can come up with a more relevant link between these two different regimes (such as in the QAHE state, the gapped surface states can somehow still contribute to the SOT?), otherwise I would suggest the authors reconsider the introduction part.

(4) The authors mentioned that the CGT can have a Curie temperature of around 80K, which is slightly higher than the Curie temperature of magnetically doped TIs. It will be great if the authors can provide the temperature-dependent SOT switching and SOT strength calibration data, to see if the TI surface contribution to SOT has the temperature dependence or not.

(5) In Fig 1c, the authors present the M_s hysteresis for the BST/CGT bilayer film. Can the authors also provide the M_s hysteresis on the pure CGT grown on the substrate? I am curious about whether the BST has any influence on the magnetic anisotropy of the CGT layer.

(6) I believe there is a typo about the Hall bar width on line 90, page 4.

Reviewer #3 (Remarks to the Author):

This is a well written article. The authors demonstrate efficient current-induced switching of the surface ferromagnetism in hetero-bilayers consisting of the topological insulator $(\text{Bi}_{1-x}\text{Sbx})_2\text{Te}_3$ and the ferromagnetic insulator $\text{Cr}_2\text{Ge}_2\text{Te}_6$, where the proximity-induced ferromagnetic surface states play two roles: efficient charge-to-spin current conversion and emergence of large anomalous Hall effect.

There are several concerns that the authors should address before the acceptance of its publication.

1. From fig 1.e, the adjacency of CGT still changes the total resistance of the device (and the change is not small if not using log scale). Does such a large resistivity change come from the Fermi level change? Will the different thickness of CGT have different impacts on the Fermi level? If so, could the 12 nm sample switching behavior be related with the Fermi level change? Also, the resistivity changing trend when adding CGT layer on BST is different at 0 and 300 K, where at 0K adding CGT will increase the resistance. Why is this impact different at 0 K and 300 K?
2. Why is the thickness in fig 2.c and 2.d different? Where is the 8 nm switching data in figure 2.c?
3. Line 131: The authors claimed that the thermal effect plays an important role in switching of the thicker samples. However, for 12 nm and 5.8 nm samples, the switching currents are similar, which means the thermal effect may not have a large difference among these two samples.
4. Line 182: it was shown in one of the PRL papers that generating SHC is most efficient when EF is at the Dirac point. Here, the authors claimed that they should observe the suppression of the charge-to-spin current conversion at DC. This conclusion should be more clarified.

Response to Reviewer # 1

Mogi and coworkers report current-induced switching of a ferromagnetic insulator in interfacial contact with a topological insulator (TI). Switching occurs presumably due to the spin-polarized surface states interacting with the magnetization of the neighboring FM insulator. They study the FM insulator layer thickness dependence and the TI composition dependence, which spans from the electron dominated to the hole dominated transport by tuning the Fermi energy. They convincingly show that this system is a very good candidate for current-induced magnetization manipulation in TI-based devices. Moreover, the systematic studies shed light on the charge-spin conversion mechanism being the surface states rather than the bulk. Overall, I believe that this is an essential work for the spintronics field and beyond. Therefore, I would recommend this work for publication in Nature Communications if the authors can address the following questions/issues.

We are grateful to the reviewer for spending precious time to review our manuscript carefully and for recognizing the value of our work. All the comments are constructive and helpful to improve our manuscript.

1) The authors quantify the switching efficiency by considering the threshold current and the coercivity of the films, together with other relevant parameters. However, based on their data, there are two potential problems with this approach. First, it ignores the magnetic anisotropy, which plays an important role in the switching process even though it is mediated by domain nucleation and propagation (see, e.g., PRB 89 214419 2014). This is because a fixed in-plane bias field will tilt the magnetization to a different angle depending on the perpendicular magnetic anisotropy. The larger the tilt angle is, the easier the switching will be. Therefore, the comparison in Fig.2e is valid only if the perpendicular anisotropy is comparable for different CGT thicknesses as the in-plane bias field was constant for all measurements. Second, the domain sizes and domain wall propagation properties will also influence the critical current density to achieve full

switching. It is evident in Fig. 2a that these properties are different for the different CGT thicknesses. Thinner CGT samples tend to have more sheared loops, evidence for small domains, and significant pinning. I would recommend the authors perform a quantitative analysis of the current-induced damping-like torque for the different samples to compare the spin-torque efficiencies. This could be easily done by, e.g., harmonic Hall voltage analysis widely used in the metallic ferromagnetic heterostructures (see, e.g., Nat. Nano. 8 587 2013, Nat Mater 12 240 2013).

Although the precise measurement of the spin-torque efficiency is not the main point in the present study, the trial for the quantitative analysis in this new system would be helpful for future studies. As suggested by the reviewer, we conducted the second-harmonic Hall voltage ($V_y^{2\omega}$) measurement in a CGT/BST bilayer. We applied a magnetic field (H_x) parallel to the ac current direction (the current amplitude $J_x = 10 \mu\text{A}$ and frequency $\omega = 13 \text{ Hz}$), where the second-harmonic Hall voltage is expected to be maximized. As shown in Fig. R1, we observed the large value of the second harmonic resistance defined by $R_{yx}^{2\omega} = V_y^{2\omega} / J_x$, which is anti-symmetric with H_x and pronounced at around the magnetic anisotropy field $H_K = 0.9 \text{ T}$. In the scenario that the second harmonic Hall resistance is generated by the spin-orbit torques, where the out-of-plane component of the magnetic oscillation produces the anomalous Hall effect. The $R_{yx}^{2\omega}$ can be given by

$$R_{yx}^{2\omega} = -\frac{1}{2} \frac{R_{yx}^{\text{AH}} H_{\text{eff}}}{|H_x| - H_K}, \quad (\text{R1})$$

where R_{yx}^{AH} stands for the out-of-plane anomalous Hall resistance. By fitting the result of Fig. R1 with this formula, the effective field H_{eff} generated by the spin-orbit torques is evaluated to be 500 mT. Furthermore, this effective field would give the extremely high charge-to-spin current conversion efficiency

$$\xi^{\text{SH}} = \frac{2e\mu_0 M_s H_{\text{eff}} t_{\text{CGT}}}{\hbar j_x} \sim 1000 \text{ nm}^{-1},$$

where $j_x = J_x/W = 10^{-2}$ A cm⁻¹ (W : the sample width 10 μ m). This value far exceeds that obtained by the current-induced switching experiments ($\xi \sim 1$ nm⁻¹) studied in the main text.

Fig. R1 | Second-harmonic Hall resistivity. In-plane magnetic field ($\mu_0 H_x$) dependence of second-harmonic Hall resistivity for the CGT($t_{\text{CGT}} = 5.8$ nm)/BST($x = 0.5$, 6 nm) with the current of 10 μ A along the magnetic field direction (x). The black line indicates the fitting curve from Eq. R1.

We have encountered such an unphysical situation also in studies on a magnetic topological insulator Cr-doped BST/BST (K. Yasuda et al., PRL **119**, 137204 (2017)) and a bulk Rashba semiconductor (Ge,Mn)Te (R. Yoshimi et al., Sci. Adv. **4**, daat9989 (2018)); To account for this, we have argued that the second-harmonic signal can appear without the spin-orbit torque induced magnetization oscillation. There, we discuss that the second harmonic Hall resistance can include another significant scattering process that the spin-momentum locked electrons are asymmetrically scattered by magnons generated in the magnetic layer. The transverse scattering under the conservation of spin-angular momentum via the magnon emission and absorption processes gives rise to a J_x^2 -proportional voltage (or second harmonic voltage) along the transverse direction. Thus, we believe that this method is inappropriate to accurately evaluate the efficiency in the present TI-based materials system. Nevertheless, such a nonlinear Hall response due to the magnon scattering can provide another concrete evidence for the spin-momentum locked surface Dirac electron dynamics. In the revised Supplemental Information VI, we have added this experimental result and

argued its inability for estimating the spin-charge conversion efficiency contrary to what is broadly anticipated in the community.

We also tried to calibrate the SOT strength in our devices by using magneto-optical Kerr microscopy (for the detail, please see the response (1) for Reviewer #2), however, we could not provide more direct experimental estimates for the spin-charge conversion efficiency, while keeping the present CGT/BST interface structure and its electronic structure intact. As stated above, this is because the spin-polarized Dirac electron dynamics is strongly coupled to the magnons excited in the proximitized magnetic layer and also because it is difficult to perform the direct magneto-optical (Kerr) probe of the magnetization dynamics of the thin *insulating* (with no plasma resonance) ferromagnetic layer (CGT). However, the main accomplishment and conclusion of this paper, i.e. that the current-induced surface Dirac electron dynamics of the topological insulator can induce the magnetization reversal in the adjacent ferromagnetic insulating layer, are valid and well supported by the experiments including these supplemented new results.

In the revised manuscript, we commented on the trial of the second harmonic measurements as follows: “Incidentally, to more quantitatively estimate the spin-charge conversion efficiency, we tried to utilize the prevailing method to use the second harmonic Hall measurements^{31,32}. The analysis based on it, however, gave an unphysically large efficiency value, i.e. $\sim 1,000 \text{ nm}^{-1}$. Such a large nonlinear Hall signal is perhaps due to a magnon scattering of the spin-momentum locked surface Dirac electrons of TI¹⁴ (see Supplementary Information VI for details).”, and described the detailed result and discussion in Supplementary Information VI.

2) Following up on the above comment, I recommend the authors report the perpendicular magnetic anisotropy estimates for the different samples used in this study.

As pointed out by the reviewer, the in-plane-field induced magnetization tilting could vary among the CGT samples with various thicknesses and accordingly could affect the switching efficiency. As suggested by the reviewer, we conducted magnetic anisotropy measurements for CGT/BST films with different CGT thicknesses (t_{CGT}) of 2.9 nm (Fig. R2a) and 5.8 nm (Fig. R2b), and also compared with the results of the CGT single layer films which were investigated in our prior publication (M. Mogi et al., APL Mater. **6**, 091104 (2018)). As shown in Fig. R2c, while both the coercive force ($\mu_0 H_c$) and anisotropy field ($\mu_0 H_K$) increase with increasing t_{CGT} , in the thin CGT layer regime ($t_{\text{CGT}} < 6$ nm) on which we mainly focus in the switching experiments, there are no drastic changes in them. Thus, the impact of magnetic anisotropy can be reasonably negligible for the present magnetization switching. In the thicker region ($t_{\text{CGT}} = 8.1$ nm and 12 nm), on the other hand, it can be a possible origin of the increase of the switching efficiency ξ as shown in Fig. 2e of the main text.

In the revised manuscript, we added the discussion and additional magnetization measurement data in the Supplementary Information III as follows.

“III. Additional magnetization data of CGT/BST bilayers.

To see if the magnetic properties of the CGT layers with various thicknesses affect the magnetization switching behaviours, we conducted magnetic anisotropy measurements for CGT/BST($x = 0.5$) films with different CGT thicknesses (t_{CGT}) of 2.9 nm (Supplementary Fig. 3a) and 5.8 nm (Supplementary Fig. 3b), and also compared with CGT single layer films which were investigated in our prior publication^{S2}. As shown in Supplementary Fig. 3c, while both the coercive force ($\mu_0 H_c$) and the anisotropy field ($\mu_0 H_K$) increase as t_{CGT} decreases, they show a saturating behaviour in the thinner CGT layer region ($t_{\text{CGT}} < 6$ nm). Thus, the impact of magnetic anisotropy change appears small for this regime. In the thicker region ($t_{\text{CGT}} = 8.1$ nm and 12 nm), on the other hand, it can be a possible origin of the increase of the switching efficiency as shown in Fig. 2e of the main text.”

[Redacted]

Fig. R2 | Dependence of CGT thickness on magnetic anisotropy. a,b, Out-of-plane (colored) and in-plane (black) magnetization hysteresis loops of the CGT($t_{\text{CGT}} = 2.9$ nm (a), 5.8 nm (b))/BST($x = 0.5$, 6 nm) heterostructures at $T = 2$ K. **c,** t_{CGT} dependence of the coercive field ($\mu_0 H_c$) (left vertical axis) and the anisotropy field ($\mu_0 H_K$) (right vertical axis). The filled and open circles indicate the CGT/BST heterostructures and the CGT single layers, where the result of the latter is adopted from M. Mogi et al., APL Mater. **6**, 091104 (2018).

3) On page 6, the authors claim that for thicker CGT samples, the reduced switching might be due to multidomain formation. This can be easily tested with an out-of-plane field and an ac current injection of the same amplitude, which would reproduce the same Joule heating scenario.

We thank the reviewer for the suggestion. We agree that if the temperature can be measured during the current injection, we can make sure that the Joule heating effect is significant for the thicker CGT samples. One way to measure it is an ac current injection as suggested by the reviewer. However, in the present experiments, we used current pulses to minimize the heating, so that we are concerned that there would be a large difference in the

sample heating effects between ac current and pulse current injections. Moreover, it is technically challenging for us to estimate the temperature in real-time during the pulse current injection. From the above concerns, we decided to use an alternative method which exploits the change of the coercive field due to current pulse injection. The details are as follows: While changing the magnetic field (H_z), where the current-induced magnetization switching does not occur, we inject current-pulse. Then, the heating shrinks the coercive field by the highest temperature reached with the current pulse injection and promotes the magnetic-field-induced magnetization reversal (Fig. R3b). We read the Hall resistance with low current excitation ($J_x = 10 \mu\text{A}$) after the sample was cooled again at each pulse. In Fig. R3c, we scale the Hall resistance coercive force H_c with the temperature (at a low current $J_x = 10 \mu\text{A}$) (Fig. R3A) and the injected current density (at the base temperature of 2 K). After all, we can estimate the temperature increase by the current pulse injection, as shown in Fig. R3d. At $j_x = 5 \text{ A cm}^{-1}$ ($J_x = 5 \text{ mA}$), which corresponds to the switching current for $t_{\text{CGT}} = 8 \text{ nm}$, 12 nm samples, the temperature gets close to $T_C (= 80 \text{ K})$ of the CGT layer. From this result, the observed reduction of the switching ratio $R_{yx}^{\text{sw}}/R_{yx}^{\text{AH}}$ therein can be ascribed to the current-induced heating effect that some part of the CGT layer reaches the temperature higher than T_C and thus cannot be reversed by the current injection.

In the revised manuscript, we describe the estimation of the heating effect in Supplementary Information IV as follows.

“IV. Temperature estimation under current pulse injection.

To estimate the heating by current-pulse injection, we use the change of the coercive field upon current pulse injection. While changing the magnetic field (H_z), we inject current-pulse, where spin-orbit torque induced magnetization switching does not occur in this magnetic field condition. Then, the heating shrinks the coercive field by the highest temperature reached with the current pulse injection and promotes the magnetic-field-induced magnetization reversal (Supplementary Fig. 4b). We read the Hall resistance with low current excitation ($J_x = 10 \mu\text{A}$) after the sample was cooled again at each pulse. In Supplementary Fig. 4c, we scale the Hall resistance coercive force H_c with the temperature (at a low current

$J_x = 10 \mu\text{A}$ (Supplementary Fig. 4a) and the injected current density (at the base temperature of 2 K). From the temperature vs. current-pulse amplitude relationship, we can estimate the temperature increase by the current pulse injection, as shown in Supplementary Fig. 4d. At $j_x \sim 5 \text{ A cm}^{-1}$ ($J_x \sim 5 \text{ mA}$), which corresponds to the switching current for $t_{\text{CGT}} = 8 \text{ nm}$, 12 nm samples, the temperature gets close to $T_C (= 80 \text{ K})$ of the CGT layer. From this result, the observed reduction of the switching ratio $R_{yx}^{\text{sw}}/R_{yx}^{\text{AH}}$, shown in Fig. 2d, can be ascribed to the current-induced heating effect that some part of the CGT layer reaches the temperature higher than T_C , and thus cannot be reversed by the current injection.”

Fig. R3 | Temperature estimation under current-pulse injection. **a,b**, Out-of-plane magnetic field ($\mu_0 H_z$) dependence of Hall resistivity under current excitation of $J_x = 10 \mu\text{A}$ measured at 2, 10, 15, 30, 40, and 60 K (a) and under current-pulses with amplitudes of $J_x = 0, 0.5, 1, 2, 3,$ and 5 mA (b) at $\mu_0 H_x = 0 \text{ T}$ in the CGT ($t_{\text{CGT}} =$

5.8 nm)/BST($x = 0.5$, 6 nm) heterostructure. **c**, Coercive field plotted as a function of temperature (bottom axis) and pulse current density (top axis). **d**, Temperature estimation as a function of the pulse current density. The broken lines indicate the switching current density for the respective-thickness CGT samples.

4) The explanation of the data of Fig.3 on page 8 is somewhat superficial (starting with; the sentence "... These features are consistent with..."). The authors should explain more clearly how they arrive at the conclusion that the TI surface state is the dominant spin current source. They should clarify in a more intuitive way why the torque direction remains the same when the carrier type changes. Also, I believe that their explanation of torques on the domain walls is oversimplified. I would suggest the authors consider a simpler picture of the macrospin model and the action of the torque on the magnetization and not on the domain walls. Although it is clear that the switching is mediated by domain nucleation and propagation, the initial domain would still need to be nucleated following a macrospin model.

We think it over that the explanation of the surface contribution to the switching was insufficient, where we only discuss the magnetization reversal direction associated with the spin-momentum locking direction. Here, we discuss that (1) the TI bulk states is less dominant in the switching, (2) the switching direction is independent of the carrier type changes, and (3) the switching direction can also be explained by a macrospin model. These additional discussions as described below and added in the revised main text can verify that the efficient current-induced magnetization switching is mediated by the topological surface state.

(1) Let us first argue the dominant source of the spin-orbit torque switching. The possibilities for the source of the spin torques other than the topological surface states may include the spin Hall effect from the bulk bands (Y. Liu et al., Nat. Commun. 9, 2492 (2018)) and the inversion symmetry breaking of the hetero-interface, where the vertical electric field at the interface can induce Rashba spin-splitting in the bulk states (F. Yang et al., Phys. Rev.

B 94, 075304 (2016)). However, these scenarios cannot account for the observation that the efficiency is increased when the E_F is within the TI bulk gap rather than at the bulk band. Therefore, the improved switching efficiency by E_F tuning (i.e., Bi/Sb composition ratio, x) comes from the surface-state dominated transport.

(2) Next, we explain why the spin torque remains the same sign against the change of carrier types of the surface state, as shown in Fig. 3c (also in Fig. R4a-c below) in the main text. The Fermi circle appears to have the opposite spin helicities for n-type and p-type (Fig. R4a), which suggests that the direction of spins would be opposite for the carrier types. However, in consideration of the Fermi surface response to the electric current (or field), the accumulated spin directions are the same as elucidated in the following. When an electric field ($+E_x$) is applied, the shift of the Fermi circle has the same direction irrespective of the carrier type due to $k_x \rightarrow k_x - \frac{eE_x\tau}{\hbar} = k_x - \delta k$. Then, as shown in Fig. R4a, if $E_F > E_{DP} = 0$, the Fermi circle shift increases the population of electrons for the $-k_x$ branch while reducing for the $+k_x$ branch. On the other hand, if $E_F < E_{DP}$, the Fermi circle shift reduces that for the $-k_x$ branch while increasing for the $+k_x$ branch. Hence, the increased spin populations for the n-type and p-type have the same direction $\sigma \parallel +y$ under $+E_x$. The antidamping effective field, which is given by $\sigma \times M$, depends not on the momentum but on the spin direction; thus, the spin-orbit torque direction is not changed by the carrier type.

Fig. R4 | **a**, Illustration of the energy dispersion of the TI surface state. **b,c**, The spin accumulation driven by a shift of the Fermi surface (top) and the difference in the Fermi distribution for electrons δf under an electric field (E_x) (bottom) for the n -type (**b**) and p -type (**c**) TI.

(3) Next, we confirm the magnetization switching direction is consistent with the spin-momentum locked surface state of BST by using the macrospin model, instead of the domain wall picture. The effective field originating from the antidamping torque $\vec{\tau}_{AD} = \tau_{AD}(\hat{m} \times (\hat{\sigma} \times \hat{m}))$, where $\tau_{AD} = \hbar J_s / (2eM_{stCGT})$ and J_s is the magnitude of the spin current, is described by $\vec{H}_{AD} = (\tau_{SO} / \mu_0) \hat{\sigma} \times \hat{m}$. Suppose the magnetic moment is $\hat{m} || + \hat{z}$, the effective field $\vec{H}_{SO} || + \hat{x}$ under the current pulse of $j_x > 0$ (i.e., $\hat{\sigma} = +\hat{y}$), which rotates the magnetization through the magnetic damping as shown in Fig. 1a in the main text. When the magnetic moment is slightly tilted to $+\hat{x}$ ($-\hat{x}$) by an in-plane magnetic field, the $\hat{m} || - \hat{z}$ ($+\hat{z}$) state is favored under the current-pulse injection, which well describes the observation shown in Fig. 2b of the main text.

In the revised manuscript, the above discussion is added to the discussion part of the main text, as follows.

“Finally, we argue that these magnetization switching features are consistent with current-induced dynamics of the spin-momentum locked TI surface state as the dominant source of the spin torques¹¹⁻¹⁹. First, possibilities for the source of the spin torques other than the topological surface states may include the spin Hall effect from the bulk bands³⁴ and the inversion symmetry breaking of the hetero-interface, where a vertical electric field at the interface can induce Rashba spin-splitting in the bulk states³⁵. However, these scenarios cannot account for the present observation that the efficiency is increased when the E_F is within the TI bulk gap rather than in the bulk states. Second, the accumulated spin direction is irrespective of the carrier types. The Fermi circle with the opposite spin helicities for n -type and p -type (Fig. 3c) suggests that the direction of spins would be opposite for the carrier types. However, in consideration of the Fermi surface response to the electric current or field, the accumulated spin directions are the same as elucidated in the following. When an electric

field ($+E_x$) is applied, the shift of the Fermi circle has the same direction irrespective of the carrier type: $k_x \rightarrow k_x - \frac{eE_x\tau}{\hbar} = k_x - \delta k$. Then, as shown in Fig. 3e, if $E_F > E_{DP} = 0$, the Fermi circle shift increases the population of electrons for the $-k_x$ branch while reducing for the $+k_x$ branch. On the other hand, if $E_F < E_{DP}$, the Fermi circle shift reduces the population of electrons for the $-k_x$ branch while increasing that for the $+k_x$ branch (Fig. 3f). Hence, the increased spin populations for the n-type and p-type have the same direction $\sigma_{||+y}$ under $+E_x$. Since the antidamping effective field, which is given by $\sigma \times M$, does not depend on the momentum but on the spin direction, the spin-orbit torque direction is not changed by the carrier types. Third, the magnetization switching direction itself is consistent with the spin-momentum locked surface state of BST by using a macrospin model²⁹: The effective field originating from antidamping torques $\vec{\tau}_{AD} = \tau_{AD}(\hat{m} \times (\hat{\sigma} \times \hat{m}))$, where $\tau_{AD} > 0$, is described by $\vec{H}_{AD} = (\tau_{AD}/\mu_0)\hat{\sigma} \times \hat{m}$. Suppose the magnetic moment is $\hat{m} || +\hat{z}$, the effective field $\vec{H}_{AD} || +\hat{x}$ under the current pulse of $j_x > 0$ (i.e., $\hat{\sigma} = +\hat{y}$ due to the spin-momentum locking nature of BST), which rotates the magnetization via magnetic damping as shown in Fig. 1a. When the magnetic moment is slightly tilted to $+\hat{x}$ ($-\hat{x}$) by an in-plane magnetic field, the $\hat{m} || -\hat{z}$ ($+\hat{z}$) state is favoured under the current-pulse injection, which well describes the observed behaviours shown in Fig. 2b. Note that once the reversed magnetic domains are nucleated by the above macrospin-model mechanism, the domains may be expanded towards the single domain state more efficiently than the macrospin rotation^{29,30}.”

5) Typo on page 4. The Hall bar width should be 10 um and not 10 mm.

We thank the reviewer for pointing out the typo.

6) The optical micrograph in Fig.1d could be improved. A higher resolution/contrast version would be better.

We replaced the previous Fig.1d with a higher resolution optical micrograph as shown also in Fig. R5 below.

Fig. R5 | Improved resolution of the optical micrograph of the Hall bar.

7) There is a misleading reference. Ref#15 reports full switching in a TI/FM conductor heterostructure. The authors have referred to <50% switching in a TI/FM insulator structure while citing this work. The discussion on page 3 should be rephrased.

We thank the reviewer for pointing out the wrong citation. We should have cited Ref. 20 (Li et al. Sci. Adv. 5, eaaw3415 (2019)), where partial spin-orbit torque switching was demonstrated in a TI/FM insulator structure.

8) The authors should estimate current density instead of stating the current itself, which depends on the device geometry.

We agree that the current density rather than the current value itself is important for deriving the efficiency. We modified the Fig. 2d in the main text (Fig. R6).

Fig. R6 | Corrected current density.

Response to Reviewer #2

In the manuscript by M. Mogi et al., the authors studied the current-induced switching and spin-orbit torque (SOT) in the topological-insulator (Bi_{1-x}Sb_x)₂Te₃/ferromagnetic-insulator Cr₂Ge₂Te₆ bilayer structure. Because of the high crystallinity and strong interfacial coupling in the bilayer, a high switching ratio (~1) is achieved. The authors also reported the Cr₂Ge₂Te₆ (CGT) layer thickness-dependent and (Bi_{1-x}Sb_x)₂Te₃ layer composition-dependent switching behaviors in this manuscript. While these results are interesting, given the abundant reports on the topological insulator (TI) -based SOT switching in the field, I don't feel this manuscript stands out as a breakthrough-type of work in this research field in terms of novelty. The manuscript should be further improved in order to be considered for the journal of Nature Communications:

We are grateful to the reviewer for spending precious time to review our manuscript carefully. We also appreciate the constructive comments, which are helpful to improve our manuscript. We responded to the comments point-by-point below.

(1) Throughout the manuscript, the authors mainly used the current-induced switching results to study the SOT in the BST/CGT bilayer structure. That is the main weakness of the work. We know that besides the switching behavior, there are several electrical-transport, magneto-optical, or microwave-induced methods to calibrate the SOT strength in the system. For example, the spin-torque ferromagnetic resonance, the hysteresis-shifting method in the presence of in-plane field and current, magneto-optical Kerr effect, etc. The authors should use one or more such methods to calibrate the SOT strength in their system, in order to have a better understanding of the SOT efficiency. The formula used by the authors (page 6, line 134) has certain constraint for describing the real SOT efficiency in the system. For example, the coercive field itself depends on many parameters, and it can also be influenced by the current applied through the film (by e.g., heating). So, this coefficient may

not be the best formula to derive the SOT efficiency. I would suggest the authors use at least one of the above-mentioned methods to carefully calibrate the SOT in the system, so that the manuscript will be more comprehensive.

Although the precise measurement of the spin-torque efficiency is not the main point in the present study, the trial for the quantitative analysis in this new system would be helpful for future studies. As one possible way to measure the SOT efficiency, we conducted a second-harmonic Hall voltage ($V_y^{2\omega}$) measurement in a CGT/BST bilayer. We applied a magnetic field (H_x) parallel to the ac current direction (the current amplitude $J_x = 10 \mu\text{A}$ and frequency $\omega = 13 \text{ Hz}$), where the second-harmonic Hall voltage is expected to be maximized. As shown in Fig. R1, we observed the large second harmonic resistance defined by $R_{yx}^{2\omega} = V_y^{2\omega} / J_x$, which is anti-symmetric against H_x and pronounced at around the magnetic anisotropy field $H_K = 0.9 \text{ T}$. In the scenario that the second harmonic Hall resistance is generated by the spin-orbit torques, where the out-of-plane component of the magnetic oscillation produces the anomalous Hall effect. The $R_{yx}^{2\omega}$ can be given by

$$R_{yx}^{2\omega} = -\frac{1}{2} \frac{R_{yx}^{\text{AH}} H_{\text{eff}}}{|H_x| - H_K}, \quad (\text{R1})$$

where the out-of-plane anomalous Hall resistance R_{yx}^{AH} . By fitting the result of Fig. R1 with this formula, the effective field H_{eff} generated by the spin-orbit torques is evaluated to be 500 mT. Furthermore, this effective field would give the extremely high charge-to-spin current conversion efficiency

$$\xi^{\text{SH}} = \frac{2e\mu_0 M_s H_{\text{eff}} t_{\text{CGT}}}{\hbar j_x} \sim 1000 \text{ nm}^{-1},$$

where $j_x = J_x / W = 10^{-2} \text{ A cm}^{-1}$ (W : the sample width $10 \mu\text{m}$). This value by far exceeds that obtained by the current-induced switching experiments ($\xi \sim 1 \text{ nm}^{-1}$) studied in the main text.

Fig. R7 | Second-harmonic Hall resistivity. In-plane magnetic field ($\mu_0 H_x$) dependence of second-harmonic Hall resistivity for the CGT($t_{\text{CGT}} = 5.8$ nm)/BST($x = 0.5$, 6 nm). The black line indicates the fitting curve from Eq. R1.

We have encountered such an unphysical situation also in studies on the magnetic topological insulator Cr-doped BST/BST (K. Yasuda et al., PRL **119**, 137204 (2017)) and the bulk Rashba semiconductor (Ge,Mn)Te (R. Yoshimi et al., Sci. Adv. **4**, daat9989 (2018)); To account for this, we have argued that the second-harmonic signal can appear without the spin-orbit torque induced magnetization oscillation. There, we discuss that the second harmonic Hall resistance can include another significant scattering process that the spin-momentum locked electrons are asymmetrically scattered by magnons generated in the magnetic layer. The transverse scattering under the conservation of spin-angular momentum via the magnon emission and absorption processes gives rise to a J_x^2 -proportional voltage (or second harmonic voltage) along the transverse direction. Thus, we believe that this method is inappropriate to accurately evaluate the efficiency in the present TI-based materials system. Nevertheless, such a nonlinear Hall response due to the magnon scattering can provide another concrete evidence for the spin-momentum locked surface Dirac electron dynamics coupled to the adjacent ferromagnetic layer. In the revised Supplemental Information, we have added this experimental result and argued its inability for estimating the spin-charge conversion efficiency contrary to what is broadly anticipated in the community.

As suggested by the reviewer, we also tried to calibrate the SOT strength in our devices by using magneto-optical Kerr microscopy. First of all, we would like to point out that the expected Kerr rotation for the CGT thin film is rather small, specifically, on the order of 0.1 mrad. or less at the lowest temperature (C. Gong et al., Nature 546, 265 (2017)), which is comparable to the conventional noise level of the laser-based Kerr microscopy in a cryostat under a relatively large magnetic field. Nevertheless, we have performed the experiments by expecting possible enhancement of the signal, such as by the near-resonant optical excitation or by the effects of heterointerfaces. We followed the setup in previous works performed on Ta/CoFeB/MgO (M. Montazeri et al., Nature Commun. 6, 8958 (2015)) and Cr-BST/BST heterostructures (X. Che et al., Adv. Mater. 32, 1907661 (2020)). Linearly polarized laser light normally incident and focused (3-5 μm spot) on the sample was chopped at ~ 100 kHz by an electro-optic modulator combined with Glan-laser prisms. The magnetization in the film was modulated by the electric current at several hundred Hz, which is detected by the double-lock-in scheme after the balanced optical detection of polarization rotation of laser light reflected on the sample surface. We checked the resolution of our setup by using several laser sources (405 nm, 532 nm, 633 nm, 690 nm), and realized that all gave a similar value of ~ 0.1 mrad., with the laser intensity less than 0.2 mW, which is exactly comparable to those achieved in the previous two works. However, unfortunately, this resolution is found to be insufficient to discuss the Kerr rotation θ_K and its current modulation $\Delta\theta_K$ in our devices with the CGT on top even at low temperatures. The θ_K seems to be less than 0.1 mrad. in our samples, due probably to the insulating nature of the CGT. We also tried several modifications on the optical setups to improve the resolution, including higher-frequency modulation of the incident laser by an acousto-optic modulator, without much success. Note that the original report on the CGT (C. Gong et al., Nature 546, 265 (2017)) employs a Sagnac interferometer to resolve small Kerr rotation, with the noise level reaching sub-micro radian, whose use is beyond our scope at this stage.

All in all, unfortunately, we could not provide more direct experimental estimates for the spin-charge conversion efficiency, while keeping the present CGT/BST interface structure and its electronic structure intact. As stated above, this is because the spin-polarized Dirac

electron dynamics is strongly coupled to the magnons excited in the proximitized magnetic layer and also because it is difficult to perform the direct magneto-optical (Kerr) probe of the magnetization dynamics of the thin *insulating* (with no plasma resonance) ferromagnetic layer (CGT). However, the main accomplishment and conclusion of this paper, i.e. that the current-induced surface Dirac electron dynamics of the topological insulator can induce the magnetization reversal in the adjacent ferromagnetic insulating layer, are valid and well supported by the experiments including these supplemented new results.

In the revised manuscript, we commented on the trial of the second harmonic measurements as follows: “Incidentally, to more quantitatively estimate the spin-charge conversion efficiency, we tried to utilize the prevailing method to use the second harmonic Hall measurements^{31,32}, which however resulted in an unphysically large efficiency value in the present case. Such a nonlinear Hall signal is perhaps due to a magnon scattering of the spin-momentum locked surface Dirac electrons of TI¹⁴ (see Supplementary Information VI for details).”, and described the detailed result and discussion in Supplementary Information VI.

(2) Another point the authors will need to address is that the studied BST/CGT bilayer structure is very close to the magnetic-TI/TI bilayer structure. We know that the Cr dopants have quite good solubility in the BST material, so during the growth of the BST/CGT bilayer, there could be possibility that the Cr elements in the CGT are diffused into the bottom BST layer. In this case, the studied BST/CGT bilayer is not much different from the previously studied magnetic-TI/TI bilayer structures. I suggest the authors to provide more evidence that the SOT is switching the CGT layer, not the diffused Cr dopants into the TI or the formed Cr-doped TI spacer layer at the interface between CGT and BST. A thorough scanning TEM and elemental mapping could be helpful in addressing this issue.

This is a fundamental issue to claim the surface-state dominant spintronic properties. In a previous study, we have investigated cross-sectional scanning TEM and EDX in our prior

publication (M. Mogi et al., PRL **123**, 016804 (2019)). In the scanning TEM, we observe a highly sharp interface without interfacial mixing between the CGT and BST layers (please see Fig. 1 of the paper). Also, in Fig. R8(b-e), we show the EDX elemental mapping in a CGT/BST/CGT heterostructure. As seen in the elemental distribution along the growth direction as averaged over the lateral direction (Fig. R8(f)), Cr and Ge atoms are well localized in the CGT layer and their amounts are almost equivalent, being consistent with the $\text{Cr}_2\text{Ge}_2\text{Te}_6$ chemical composition, while Bi atoms locate only in the BST layer. Furthermore, we used X-ray and polarized neutron reflectometry on the same heterostructure (please see Fig. 2). These macroscopic measurements strongly support the microscopic TEM/EDX results. Thus, the spin-orbit torques directly influence the magnetization of the CGT layer.

In the revised manuscript, we revised the main text to describe the previous study: “The previous study²⁵ on the CGT/BST heterostructure, that was prepared in the same way, has proven the high crystal quality and well-ordered, sharp interfaces due to van der Waals bonding as well as negligible atomic interdiffusion by cooperatively using x-ray diffraction, depth-sensitive x-ray/neutron reflectometry, and cross-sectional scanning transmission electron microscopy/energy-dispersive x-ray spectroscopy (Supplementary Information I).”, and added Supplementary Information I to show the TEM/EDX mapping.

Fig. R8 | TEM/EDX measurements in a CGT/BST/CGT heterostructure. (a) Scanning TEM image corresponding to the EDX scan area. (b-e) Elemental distribution maps for each element, Cr (b), Ge (c), Bi (d), and Te (e). (f) Line profiles of Cr, Ge, Bi, and Te. Figures are adopted from M. Mogi et al, PRL **123**, 016804 (2019).

(3) In the first two paragraphs, or the introduction part, the authors have addressed that the BST/CGT could be useful in studying the quantum anomalous Hall effect (QAHE), axion insulators, and SOT. While these objects all involved the topological materials, the mechanisms are quite different. For example, in the SOT study, the TI surface states are conducting while in the QAHE, the TI surface is gapped and only the edge states participate in the electrical transport. It will be good if the authors can come up with a more relevant link between these two different regimes (such as in the QAHE state, the gapped surface states can somehow still contribute to the SOT?), otherwise I would suggest the authors reconsider the introduction part.

We would like to add more explanation about the link between the current-induced magnetization switching and QAHE. As pointed out by the reviewer, since only the edge state carries their transport at the equilibrium condition of the QAH state, it cannot contribute to the anti-damping torque magnetization switching. Here we propose possible two routes to connect the two different physics. One route is electrostatic gating, where the Fermi level can be effectively shifted from an insulating state to a surface conducting state. Another route is the current-induced instability of the QAH state. If a large lateral current ($j_x > 50 \mu\text{A}/\text{cm}$) is applied across the sample, the QAH state breaks down because the large electric field generates holes and electrons from the gapped surface state [M. Kawamura et al., PRL **119**, 016803 (2017), PRB **102**, 041301(R) (2020), E. J. Fox et al., PRB **98**, 075145 (2018)]. Then, the nonzero longitudinal conductivity is observed as a signature of the surface state transport presumably with spin-momentum locking. Given the fact that the carrier type of the surface state does not affect the magnetization switching direction as also clarified in the present

study, one can expect the magnetization switching is possible in the QAH state under strong current pulse injection.

In the revised manuscript, we have added the above ideas to the conclusion part of the main text as follows: “Whereas such topological states are surface insulating and would not directly contribute to the spin-torque generation, an additional electrostatic gating capability to control the Fermi level or a current-driven breakdown of the QAH state during a current-pulse injection³⁶⁻³⁸ could retrieve the spin-polarized surface transport”.

(4) The authors mentioned that the CGT can have a Curie temperature of around 80K, which is slightly higher than the Curie temperature of magnetically doped TIs. It will be great if the authors can provide the temperature-dependent SOT switching and SOT strength calibration data, to see if the TI surface contribution to SOT has the temperature dependence or not.

Whereas we could not find a way to accurately calibrate the SOT efficiency (please see the comment (1)), we conducted the temperature-dependent SOT switching experiments in the CGT($t_{\text{CGT}} = 5.8$ nm)/BST($x = 0.5$, 6 nm), in which finite spontaneous magnetization is seen up to 50 K (Fig. R9a). From the results of the current-induced switching experiments (Fig. R9b) and the magnetization measurements (Fig. R9c), we extract the switching efficiency coefficient ξ [$= 2e\mu_0 M_s H_c t_{\text{CGT}} / (\hbar j_x^{\text{sw}})$] and switching ratio $R_{yx}^{\text{sw}}/R_{yx}^{\text{AH}}$ (Fig. R9d). While the switching is accomplished at all the temperatures below 50 K as seen in the nearly constant $R_{yx}^{\text{sw}}/R_{yx}^{\text{AH}}$, the ξ sharply increases as decreasing the temperature. We speculate that the enhancement of ξ is due to the enhancement of proximity coupling which is evident in the sharp increase of the anomalous Hall resistance (R_{yx}^{AH}) shown in Fig. R9c, rather than the moderate increase of the spontaneous magnetization (M_s). The enhanced proximity coupling makes the coupling between the surface state spins and localized spins strong, and could result in the more efficient spin-orbit torque switching.

In the revised manuscript, we have added the temperature-dependent switching in Supplementary Information VII as follows.

“VII. Temperature dependence of current-induced magnetization switching.

We conducted the temperature-dependent current-induced switching experiments in the CGT($t_{\text{CGT}} = 5.8$ nm)/BST($x = 0.5$, 6 nm), in which finite spontaneous magnetization is seen up to 50 K (Supplementary Fig. 7a). From the results of the current-induced switching experiments (Supplementary Fig. 7b) and the magnetization measurements (Supplementary Fig. 7c), we extract the switching efficiency coefficient ξ [$= 2e\mu_0 M_s H_c t_{\text{CGT}} / (\hbar j_x^{\text{sw}})$] and the switching ratio $R_{yx}^{\text{sw}}/R_{yx}^{\text{AH}}$ (Supplementary Fig. 7d). While the switching is accomplished at all the temperatures below 50 K as seen in the nearly constant $R_{yx}^{\text{sw}}/R_{yx}^{\text{AH}}$, the ξ sharply increases with decreasing the temperature. We speculate that the enhancement of ξ is due to the enhancement of proximity coupling which is evident in the sharp increase of the anomalous Hall resistance (R_{yx}^{AH}) shown in Supplementary Fig. 7c, rather than the moderate increase of the spontaneous magnetization (M_s). The enhanced proximity coupling makes the coupling between the surface state spins and localized spins strong, and could result in the more efficient spin-orbit torque switching.”

Fig. R9 | Dependence of temperature on magnetization switching. **a**, $\mu_0 H_z$ dependence of the Hall resistance R_{yx} in the CGT($t_{\text{CGT}} = 5.8$ nm)/BST($x = 0.5$, 6 nm) at various temperatures ($T = 2, 10, 20, 30, 50$, and 70 K). **b**, Magnetization switching in the CGT/BST devices at various temperatures ($T = 2, 10, 20, 30$, and 50 K) under $\mu_0 H_x = -0.1$ T. **c**, T dependence of the spontaneous anomalous Hall resistance R_{yx}^{AH} and spontaneous magnetization M_s . **d**, T dependence of the coefficient ξ [= $2e\mu_0 M_s H_c t_{\text{CGT}} / (\hbar j_x^{\text{sw}})$] (left axis) and the switching ratio of $R_{yx}^{\text{sw}}/R_{yx}^{\text{AH}}$ (right axis).

(5) In Fig 1c, the authors present the M_s hysteresis for the BST/CGT bilayer film. Can the authors also provide the M_s hysteresis on the pure CGT grown on the substrate? I

am curious about whether the BST has any influence on the magnetic anisotropy of the CGT layer.

In our prior publication (M. Mogi et al., *APL Mater.* **6**, 091104 (2018)), we have shown that the BST layers do not affect the magnetic properties of the CGT layers. In the study, we grew a CGT layer on a 2 nm BST buffer layer (Fig. R10(a)) and without any buffer layer (Fig. R10(b)) on InP substrates. As shown in Fig. R10, we observed that hysteresis curves with perpendicular magnetic remanence and anisotropy fields obtained by in-plane magnetization measurements are consistent in both films.

In the revised manuscript, we added the above comments in the main text as follows: “The magnetization of a MBE-grown CGT (12 nm)/BST(6 nm) structure (the Curie temperature: $T_C \sim 80$ K) at a temperature of $T = 2$ K (Fig. 1c) shows the hysteresis loop with the out-of-plane easy axis, which is nearly identical with the property of the MBE-grown CGT single layer itself²⁴.”

[Redacted]

Fig. R10 | Impact of BST layers on magnetic properties of CGT. Magnetization (M) curves normalized by the saturation magnetization (M_s) for 36-nm-thick $\text{Cr}_2\text{Ge}_2\text{Te}_6$ films with a 2 QL $(\text{Bi,Sb})_2\text{Te}_3$ buffer layer (a) and without $(\text{Bi,Sb})_2\text{Te}_3$ buffer layer (b) grown under the identical condition of $P_{\text{Ge}}/P_{\text{Cr}} = 3.2$. Figures are adopted from the reference [M. Mogi et al., *APL Mater.* **6**, 091104 (2018) (FIG. S2)].

(6) I believe there is a typo about the Hall bar width on line 90, page 4.

We thank the reviewer for pointing out the typo. We corrected it to 10 μm .

Response to Reviewer #3

This is a well written article. The authors demonstrate efficient current-induced switching of the surface ferromagnetism in hetero-bilayers consisting of the topological insulator $(\text{Bi}_{1-x}\text{Sbx})_2\text{Te}_3$ and the ferromagnetic insulator $\text{Cr}_2\text{Ge}_2\text{Te}_6$, where the proximity-induced ferromagnetic surface states play two roles: efficient charge-to-spin current conversion and emergence of large anomalous Hall effect. There are several concerns that the authors should address before the acceptance of its publication.

We would like to thank the reviewer for spending precious time for the review and appreciate the recognition of the significance of this work. All the comments are constructive and helpful to improve our manuscript.

1. From fig 1.e, the adjacence of CGT still changes the total resistance of the device (and the change is not small if not using log scale). Does such a large resistivity change come from the Fermi level change? Will the different thickness of CGT have different impacts on the Fermi level? If so, could the 12 nm sample switching behavior be related with the Fermi level change? Also, the resistivity changing trend when adding CGT layer on BST is different at 0 and 300 K, where at 0K adding CGT will increase the resistance. Why is this impact different at 0 K and 300 K?

We thank the reviewer for the sharp questions. To clarify the impact of the CGT layer adjacent to the BST layer, we show in Fig. R11 the temperature dependence of sheet resistance R_{xx} of the CGT/BST bilayers with various CGT thicknesses and the BST single layer. First, the 12 nm sample is not so peculiar as compared with the thinner thickness CGT samples, so that the switching behavior of the 12 nm sample (Fig. 2 in the main text) is not relevant to the possible Fermi level changes with the change of the CGT layer thickness. Next, as also pointed by the reviewer, the resistivity of CGT/BST samples shows an insulating behavior while the BST single layer exhibits a metallic behavior. We would raise

two possible reasons: (1) holes are slightly transferred from the adjacent CGT layer to the BST layer and then the chemical potential approaches the Dirac point since the chemical potential of the BST single layer (Sb fraction $x = 0.5$) lies above the Dirac point, and (2) the exchange interaction between CGT and BST layers introduces the magnetic gap in the Dirac surface state, which removes the weak antilocalization property of the gapless Dirac surface state.

In the revised manuscript, we show Fig. R11 and add the comments described above in Supplementary Information II as follows:

“We investigated the CGT thickness (t_{CGT}) dependence of transport in CGT/BST ($x = 0.5$) bilayers. As shown in Supplementary Fig. 2a, the change of t_{CGT} does not affect their sheet resistance R_{xx} , which may rule out possible effects of electronic structure changes in the bilayers on the thickness dependence of the magnetization switching (Fig. 2 in the main text). In addition, we compare R_{xx} of the CGT/BST bilayers with that of the BST single layer (Supplementary Fig. 2b). They show insulating behaviours while the BST single layer exhibits a metallic behaviour. We raise two possible reasons for the different temperature trends: (1) Holes are slightly transferred from the adjacent CGT layer to the BST layer and then the chemical potential approaches the Dirac point since the chemical potential of the BST single layer (Sb fraction $x = 0.5$) lies above the Dirac point, and (2) the exchange interaction between the CGT and BST layers opens the magnetic gap in the Dirac surface state, which removes the antilocalization property of the gapless Dirac surface state.”

Fig. R11 | T dependence of R_{xx} in CGT/ $(\text{Bi}_{1-x}\text{Sb}_x)_2\text{Te}_3$ (6 nm) bilayer films with various CGT thickness (t_{CGT}).

2. Why is the thickness in fig 2.c and 2.d different? Where is the 8 nm switching data in figure 2.c?

Following this reviewer's comment, we added the 8 nm switching data as shown in Fig. R12.

Fig. R12 | Magnetization switching in the CGT/BST devices with various t_{CGT} (= 2.9, 4.6, 5.8, 8.1, and 12 nm) under $\mu_0 H_x = -0.1$ T.

3. Line 131: The authors claimed that the thermal effect plays an important role in switching of the thicker samples. However, for 12 nm and 5.8 nm samples, the switching currents are similar, which means the thermal effect may not have a large difference among these two samples.

To look into the reason why the switching current seems to be almost the same for 12 nm and 8.1 nm, we estimate the heating by comparing the coercive field between the low-current excitation transport measurement and the current-pulse induced measurement where the heating shrinks the coercive field by the highest temperature reached with the current pulse injection (Fig. R13b). We read the Hall resistance with low current excitation ($J_x = 10 \mu\text{A}$) after the sample was cooled again at each pulse. In Fig. R13c, we scale the Hall resistance coercive force H_c with the temperature (at a low current $J_x = 10 \mu\text{A}$) (Fig. R13A) and the injected current density (at the base temperature of 2 K). After all, we can estimate the temperature increase by the current pulse injection, as shown in Fig. R13d. At $j_x = 5 \text{ A cm}^{-1}$ ($J_x = 5 \text{ mA}$), which corresponds to the switching current for $t_{\text{CGT}} = 8 \text{ nm}$, 12 nm samples, the temperature gets close to $T_C (= 80 \text{ K})$ of the CGT layer. From this result, we interpret that the temperature of some part of the CGT layer exceeds the T_C and hence that the heating would contribute to the switching efficiency values.

Fig. R13 | Temperature estimation under current-pulse injection. **a,b,** Out-of-plane magnetic field dependence of Hall resistivity under current excitation of $J_x = 10 \mu\text{A}$ measured at 2, 10, 15, 30, 40, and 60 K (a) and under current-pulses with amplitudes of $J_x = 0, 0.5, 1, 2, 3,$ and 5 mA (b) at $\mu_0 H_x = 0 \text{ T}$ in the CGT($t_{\text{CGT}} = 5.8 \text{ nm}$)/BST($x = 0.5, 6 \text{ nm}$) heterostructure. **c,** Coercive field plotted as a function of temperature (bottom axis) and pulse current density (top axis). **d,** Temperature estimation as a function of the pulse current density. The broken lines indicate the switching current density for the respective-thickness CGT samples.

4. Line 182: it was shown in one of the PRL papers that generating SHC is most efficient when EF is at the Dirac point. Here, the authors claimed that they should observe the

suppression of the charge-to-spin current conversion at DC. This conclusion should be more clarified.

As the reviewer points out, there is a possibility that the in-plane spin-polarization can be maximized when the Fermi level locates near the Dirac point [H. Wu et al., PRL 123, 207205 (2019)]. Also, there is a controversial study [Kondou et al., Nat. Phys. 12, 1027 (2016)] that the charge-to-spin current conversion efficiency can be lowered by taking into account the reduction of spin-polarization near the Dirac point as observed by spin-polarized ARPES. In our study, however, the Dirac point is already gapped out due to the existence of strong proximity coupling between the topological surface state and the ferromagnetic insulator layer, in contrast to the previous studies which use a non-magnetic metal insertion layer (e.g., Ti or Cu) between the ferromagnet and TI layers. In the present case, when the Fermi level is close to the magnetic gap, the in-plane spin-polarization would be weaker, thereby potentially reducing the charge-to-spin current conversion efficiency. Nonetheless, the result of this study is that we did not observe any clear suppression of the efficiency, possibly due to still unprecise tuning of the Fermi level into the middle of the magnetic gap or to the spatially inhomogeneous gap opening.

In the revised manuscript, we discussed the reason for the potential suppression of spin-to-charge conversion in our system in the main text as follows: “Note that despite the surface state is magnetically gapped by the proximity coupling with the FMI layer, suppression of the charge-to-spin current conversion efficiency in the nearly charge-neutral samples is not observed, possibly due to still unprecise tuning of the Fermi level into the middle of the exchange gap or to the spatially inhomogeneous gap opening.”

REVIEWERS' COMMENTS

Reviewer #1 (Remarks to the Author):

The authors have satisfactorily addressed all of my comments and questions; therefore, I now recommend this work for publication in Nature Communication.

Reviewer #2 (Remarks to the Author):

I believe the authors have answered my questions in a satisfactory manner and they have improved the manuscript accordingly. I think the manuscript is now suitable for publication in Nature Communications.

Reviewer #3 (Remarks to the Author):

The authors have addressed most of my concerns well with a complete supporting information.

There are two more minor concerns for the authors to further improve their manuscript. 1. There are some good and relevant references on study of switching and reading of Topological insulator / Ferromagnetic insulator bilayer and devices uploaded in arXiv. The authors should consider to put them in the references. 2. The Fig. R2 (supporting section), could the authors add more data points beyond just 2K.

In summary, I recommend its publication after those two minor revisions.

Response to Reviewer # 1

The authors have satisfactorily addressed all of my comments and questions; therefore, I now recommend this work for publication in Nature Communication.

We thank the reviewer for many valuable comments to improve our manuscript and for the recommendation for the publication in Nature Communications.

Response to Reviewer #2

I believe the authors have answered my questions in a satisfactory manner and they have improved the manuscript accordingly. I think the manuscript is now suitable for publication in Nature Communications.

We thank the reviewer for many valuable comments to improve our manuscript and for the recommendation for the publication in Nature Communications.

Response to Reviewer #3

The authors have addressed most of my concerns well with a complete supporting information.

There are two more minor concerns for the authors to further improve their manuscript. 1. There are some good and relevant references on study of switching and reading of Topological insulator / Ferromagnetic insulator bilayer and devices uploaded in arXiv. The authors should consider to put them in the references. 2. The Fig. R2 (supporting section), could the authors add more data points beyond just 2K.

In summary, I recommend its publication after those two minor revisions.

We thank the reviewer for many valuable comments to improve our manuscript and for the recommendation for the publication in Nature Communications.

To respond to the reviewer's suggestions, (1) we searched papers recently posted on arXiv, then we found one relevant, good paper, of which the work was done by Chiba and Komine although it has already been published in Phys. Rev. Applied. We hope that this is the one that the reviewer implies. Also, (2) we show the data of the temperature dependence of the perpendicular magnetic anisotropy in Supplementary Information as also shown below.

Supplementary Fig. 4 | Dependence of temperature on magnetic anisotropy. a,b, Out-of-plane (a) and in-plane (b) magnetization hysteresis loops of the CGT($t_{\text{CGT}} = 5.8$ nm)/BST($x = 0.5$, 6 nm) heterostructures at various temperatures (T). **c,** T dependence of the coercive field ($\mu_0 H_c$) (left axis) and the anisotropy field ($\mu_0 H_K$) (right axis). The error bars indicate measurement uncertainty.